# Red Meat Heating Processes, Toxic Compounds Production and Nutritional Parameters Changes: What about Risk–Benefit?

**DOI:** 10.3390/foods13030445

**Published:** 2024-01-30

**Authors:** Marco Iammarino, Rosaria Marino, Valeria Nardelli, Mariateresa Ingegno, Marzia Albenzio

**Affiliations:** 1Department of Chemistry, Istituto Zooprofilattico Sperimentale della Puglia e della Basilicata, 71121 Foggia, Italy; valeria.nardelli@izspb.it (V.N.); mariateresa.ingegno@izspb.it (M.I.); 2Department of Agriculture, Food, Natural Resources and Engineering (DAFNE), University of Foggia, 71121 Foggia, Italy; marzia.albenzio@unifg.it

**Keywords:** acrylamide, heated red meat, heterocyclic aromatic amines, nitrosamines, nutritional value, polycyclic aromatic hydrocarbons

## Abstract

The heating process is a crucial step that can lead to the formation of several harmful chemical compounds in red meat such as heterocyclic aromatic amines, N-Nitrosamines, polycyclic aromatic hydrocarbons and acrylamide. Meat has high nutritional value, providing essential amino acids, bioactive compounds and several important micronutrients which can also be affected by heating processes. This review aims to provide an updated overview of the effects of different heating processes on both the safety and nutritional parameters of cooked red meat. The most-used heating processes practices were taken into consideration in order to develop a risk–benefit scenario for each type of heating process and red meat.

## 1. Introduction

Meat heating processes could result in a number of harmful chemical compounds, so the World Health Organization (WHO) and the Food and Agriculture Organization (FAO) of the United Nations discourage high consumption of meat, especially red meat. This suggestion is due to some toxic effects on humans, such as cardiovascular diseases and cancer risks that these molecules may exercise, especially after heating processes. In particular, the International Agency for Research on Cancer (IARC) Working Group classified processed meat as “carcinogenic to humans” (Group 1), on the basis of sufficient evidence for colorectal cancer, and red meat as “probably carcinogenic to humans” (Group 2A), based on substantial evidence of a positive association between consumption and colorectal cancer, other than strong mechanistic evidence [1]. Despite these aspects, meat consumption is increasing worldwide, particularly in developing countries [2].

On the other hand, meat has high nutritional value, providing essential amino acids, bioactive compounds and several important micronutrients such as iron, zinc and vitamins B3 and B12 [3]. The heating process is a key phase for meat, since it makes foods microbiologically safe to consume and palatable; however, heating processes affect the nutritional value of meat due to changes in some components [4]. It is an extremely complex subject that includes many variables often related to each other.

Several chemical compounds have been listed as possible toxic contaminants of meat. These compounds may be present at low levels in raw meat, but the concentration may increase after heating processes and if preservative agents, such as nitrite/nitrate, have been added to the product [5]. The Heterocyclic Aromatic Amines (HAAs), N-Nitrosamines (NAs), Polycyclic Aromatic Hydrocarbons (PAHs) and Acrylamide have been indicated in the scientific literature as groups of chemicals of concern in meat [6]. Most of these compounds are listed among “processing contaminants” since their levels in the product depend on the specific heating treatment [7] and/or on other food processing. The occurrence of these classes of compounds in meat has been investigated during the last few years, especially for PAHs [8]. However, many aspects of food safety are still worthy of research. Among these aspects, the evaluation of the contemporary presence of different toxic compounds, also at very low levels, the so-called “cocktail effect”, represents a new frontier in food safety knowledge. In 2019, the European Food Safety Authority (EFSA) released a scientific report focused on understanding chemical mixtures in food, highlighting that the consumer’s level of concern from being exposed to the combined effects of chemicals in food is high [9]. It is worthy of mentioning that there are many gaps in knowledge and, taking into account the significant toxic effects of HAAs, NAs, PAHs and acrylamide described above, and their possible simultaneous presence in meat, further insights are needed.

Another important aspect of food safety, related to meat consumption, is the high variability of toxic compound levels due to the specific product composition and heating processes. Indeed, the differences in meat composition, especially between fresh and processed meat, and relating to the specific lipid/protein profile of meat from different species, influence the formation of HAAs, NAs, PAHs and acrylamide significantly. Previous studies [10,11,12] on the effects of product type/heat treatment type highlighted a correlation with toxic compound composition; this finding is crucial for food safety and it is useful for improving meat product formulations and for correct heat treatment management.

To date, the available literature has discussed the effects of cooking on meat from different animal species such as cattle, chicken or pig (e.g., [13,14,15]), but on the basis of our knowledge, the effect of thermal processing on red meat from different animal species such as sheep, goat and horse, focusing on toxic compounds production and nutritional parameters changes, has not been reviewed.

This review provides an updated overview of the effects of different heating processes, such as grilling, roasting/oven-broiling, boiling, microwaving, frying and sous vide, on toxic compounds production, alone or simultaneously, and on the nutritional parameters of red meat from different animal species (cattle, pig, sheep, goat and horse).

The purpose of this review is to consolidate all the information to critically assess the risks and benefits associated with the heating processes of meat from different animal species. Strategies to minimize the risks associated with meat consumption and to preserve meat’s beneficial nutritional properties have also been reported.

## 2. Effect of Heating Processes on Meat Safety

The most significant effects of heating processes on the formation of different contaminants in meat from different animal species have been reported in Table 1.

### 2.1. Heterocyclic Aromatic Amines

HAAs are formed from the reaction between creatine/creatinine, amino acids and sugars. All precursors are present at different levels in meat and the reaction increases with the temperature and heating process duration [37]. Among different HAAs, 2-Amino-1-methyl-6-phenylimidazo[4,5-b] pyridine (PhIP) and 2-Amino-3,8-dimethylimidazo[4,5-f] quinoxaline (MeIQx) are the most abundant in cooked meat and most absorbed [38,39]. A tentative full list of HAAs is composed of 30 compounds [35] including 2-Amino-3,4,8-trimethylimidazo[4,5-f] quinoxaline (4,8-DiMelQx), 9H-pyrido [4,3-b] indole (Norharman) and 1-methyl-9H-pyrido[4,3-b] indole (Harman) as other important HAAs. The concentration of precursors is highly variable among meat from different animal species [16]. This characteristic, together with the variability due to several types of heating processes used for meat preparation, leads to high differences in HAA levels in cooked meat. As an example, the reaction of the formation of PhIP is schematized in Figure 1 [35,40]. Taking into account the most representative literature available on the topic, it is possible to extrapolate several data in order to evaluate how different types of heating process treatment can influence HAA formation in meat from different animal species. For instance, considering beef patty samples, and the sum of five HAAs (namely, PhIP, MeIQx, 4,8-DiMeIQx, Norharman and Harman), it is possible to compare three types of meat heating processes, characterized by comparable combinations of time/temperature (198–200 °C—12 min for frying/grilling and 191 °C—20 min for oven-broiling). A total amount of 10.5 µg kg^−1^ of HAAs was detected in fried samples by Thiebaud et al. [20], and this value was about double that detected by grilling at 200 °C and 240 °C (5.1 µg kg^−1^) [17] and about ten times the total concentration detected after oven-broiling (1.62 µg kg^−1^) [18]. 4,8-DiMeIQx, the compound that demonstrated the highest carcinogenic potencies among HAAs, showed the highest concentration (1.3 µg kg^−1^) after frying [41], while the concentration detected after oven-broiling was 0.02 µg kg^−1^ and a “no detectable” result was obtained after grilling.

The same comparison can be made for pork patty samples, substantially obtaining the same findings. Indeed, overall HAAs concentrations (the sum of PhIP, MeIQx, 4,8-DiMeIQx) equal to 16.7 and 5.0 µg kg^−1^ were quantified by Shin [34] after frying and oven-broiling, respectively. These authors used a Teflon-coated electric pan, temperatures in the range of 177–225 °C and heating processes times in the range of 8–19 min. After grilling, using a double contact grill at 230 °C, the sum of all five HAAs quantified was equal to 5.5 µg kg^−1^ [16]. Shin [34] also tested boiling (T: 100 °C, time: 8–16 min), verifying a significant reduction of HAAs formation, only detecting MeIQx at a concentration in the range 0.4–1.0 µg kg^−1^.

On the basis of the above-mentioned, it is possible to affirm that frying is the heating process that causes the highest formation of HAAs in cattle and pig meat compared to grilling and oven-broiling. A complete view in decreasing order of HAAs formation can be schematized as follows: frying > grilling > oven-broiling > boiling.

### 2.2. N-Nitrosamines

Among N-nitroso compounds, chemicals involved in the development of cancer in humans and animals, NAs are the most important class in food, due to the availability of precursors both as natural compounds and food additives (nitrite) [42]. The mechanism of formation of NAs consists of reactions between nitrosating agents (NOX) and amino-based substances (R_2_NH):R_2_NH + NOX ↔ R_2_NNO + HX

NAs are considered pro-carcinogenic, which means that they have to be activated. Metabolic activation leads to the formation of DNA adducts, which are critical for their mutagenic and carcinogenic activity. N-Nitrosodimethylamine (NDMA) is the simplest NA and the most detected in food. Its involvement in hepatic cancer has been extensively studied [43]. The critical step for its carcinogenicity is the α-hydroxylation of the methyl group. This reaction is often catalyzed by cytochrome P450 2E1 and the product is α-hydroxydimethylnitrosamine, which is highly unstable and rapidly decomposes to form formaldehyde, methyl diazohydroxide and/or methyl diazonium ion. The reactions of the formation of NDMA and its subsequent activation can be schematized as shown in Figure 2.

The role of heating processes on NA formation in meat is still to be fully defined. Indeed, although high temperatures can promote the reaction between nitric oxide and secondary amines, the marked volatility of several NAs could lead to their decrease after heating processes. Moreover, most studies available in the literature are focused on processed meat (added with nitrite and nitrate) and few data are available regarding fresh meat [44].

However, a specific study carried out by Yurchenko and Molder [23] demonstrated the substantial absence of five important NAs in cattle, pig and mutton raw meat, and their formation after smoking, grilling and frying. From this study, it is possible to compare the overall formation of five NAs obtained after the grilling and frying of pork meat, corresponding to 9.7 and 13.3 µg kg^−1^, respectively. The greatest risk of frying, if compared to grilling, is also confirmed considering that the concentration of the most potent carcinogen among the five NAs, N-Nitrosodimethylamine, in fried pork meat resulted in twice (0.66 µg kg^−1^) that formed after grilling (0.34 µg kg^−1^).

Thus, the results available regarding NA formation in meat after heating processes substantially confirm that the increase obtained after frying is higher than that caused by grilling.

On a global scale, there is no uniformity regarding the legal limits of NAs in food. The United States Department of Agriculture (USDA) established a limit of 10 µg kg^−1^ for total volatile NAs in cured meat products, while China and Chile defined a limit only for food N-Nitrosodimethylamine, equal to 4 and 7 µg kg^−1^ in fish and related products and to 30 µg kg^−1^ in cured meat products in China and Chile, respectively. Estonia and the Russian Federation established a limit for the sum of N-Nitrosodimethylamine and N-Nitrosoethylamine (equal to 2 and 4 µg kg^−1^ in meat sausages submitted to heat treatment and smoked meat sausages, respectively). Finally, Canada defined a legal limit equal to 10 µg kg^−1^ for the sum of N-Nitrosodimethylamine, N-Nitrosoethylamine, N-Nitrosodipropylamine, N-Nitrosodibutylamine, N-Nitrosopiperidine and N-Nitrosomorpholine, and to 15 µg kg^−1^ for N-Nitrosopyrrolidine, both in cured meats [45]. No limit has been established in Europe so far. Looking at the regulations established worldwide and the EFSA Opinion on NAs published in 2023 [46], a quick definition of legal limits for at least N-Nitrosodimethylamine and N-Nitrosoethylamine in meat products seems urgent.

### 2.3. Polycyclic Aromatic Hydrocarbons

PAHs can be present in meat as environmental contaminants and/or as a consequence of processing and heating processes (processing contaminants). Compared to the previous categories, the levels of these compounds are monitored by health authorities, especially in developed countries, [47] and the European Commission established the maximum levels for PAHs in foodstuffs in Regulation (EU) No. 2023/915 [48]. In October 2005, the IARC evaluated the carcinogenicity of 60 PAHs. The Working Group concluded that benzo-(a)-pyrene (BaP) is the most investigated compound, classified as carcinogenic to humans (IARC group 1). As previously described for NAs, BaP is also metabolically activated to form a diol epoxide derivative (Figure 3). Cyclopenta(c,d)pyrene, dibenz(a,h)anthracene and dibenzo(a,l)pyrene were classified as probably carcinogenic to humans (group 2A), while benz(a)anthracene, benzo(b)fluoranthene, benzo(j)fluoranthene, benzo(k)fluoranthene, chrysene,dibenzo(a,h)pyrene, dibenzo(a,i)pyrene,indeno(1,2,3-cd)pyrene, 5-methyl-chrysene, benz(j)acean-thrylene and benzo(c)phenanthrene were listed among possibly carcinogenic substances (group 2B). Finally, it is worth mentioning that some PAHs, particularly dibenzo(a,l)pyrene, seem to be more potent carcinogens than benzo(a)pyrene [49]. A possible mechanism of formation of benzo-(a)-pyrene is shown in Figure 3 [50].

PAH levels can increase depending on the heating process procedure as a consequence of incomplete combustion of charcoal, pyrolysis of organic materials and the direct contact of flame/fat [47]. Regarding meat, more than 30 PAHs have been identified, comprising benzo[a]pyrene. It is also important to underline that the IARC Working Group concluded that data about PAH levels in food are limited, and large-scale, independent cohort studies are needed [49]. This is also confirmed by looking at the studies focused on the effect of different types of heating processes on the formation of PAHs in meat [24,51]. Within the few studies available on red meat, Sahin et al. [25] detected an overall number of PAHs (15 compounds) equal to 4.45 µg kg^−1^ in grilled beef and sheep meatballs, with a specific content of BaP of 0.7 µg kg^−1^. The authors used electric and charcoal grilling (both direct) and did not test other types of heating processes. Sumer and Oz [26] studied the effect of direct and indirect barbecuing when heat processing meat. The sum of eight PAHs after direct grilling results in almost thrice (11.3 µg kg^−1^) the amount quantified after indirect grilling (4.4 µg kg^−1^), with comparable concentrations of BaP (0.49 and 0.39 µg kg^−1^ after direct and indirect grilling, respectively). The authors only studied this type of heating process in this case. A study where a different types of heating processes were tested to verify PAH formation was published by Djinovic et al. [52]. In this study, beef and pork ham samples were smoked, after packing and storage, and the presence of 16 PAHs was verified. Benzo[c]fluorene was the PAH detected at the highest level both in beef and pork (up to 2.1 μg kg^−1^).

However, the study conducted by Siddique et al. [53] on rabbit meat confirms what was stated above, that frying has the highest impact on contaminants production compared to grilling, microwaving and oven roasting, with the lowest impact obtained after boiling. Indeed, this study reported fluorene contents up to 0.13 and 0.02 μg kg^−1^ after frying and boiling treatment, respectively.

### 2.4. Acrylamide

It is well known that acrylamide is a toxic compound, produced in food during heating processes such as frying, baking and roasting. In the human body, it decomposes forming glacidamide which can cause DNA mutation and damage to the nervous system [30]. The IARC classified acrylamide as probable carcinogenic to humans (group 2A). This compound derives from the Maillard reaction [54]. A possible mechanism of formation is schematized in Figure 4 [55].

The presence of quantifiable levels of acrylamide was demonstrated, other than in starch-rich foods, in meat after heating processes [30,54]. Thus, this compound adds to other toxic substances listed above, composing a general picture that arouses much concern in food safety.

Few data are available regarding acrylamide formation in meat after heating processes relying on frying treatment. The authors reported that the combination of time/temperature affects the formation of this contaminant, while the oil type has no impact [30,54]. Basaran and Faiz [56] recently published other data related to acrylamide formation in several traditional meat products in India. From this study, it is possible to obtain interesting information that confirms what is stated above relating to the effect of different heating processes on the final level of contaminants in meat. Indeed, these authors reported that meat products composed of a mixture of cattle and lamb meat can be characterized by different levels of acrylamide after heating processes by grilling, oven-broiling and frying. The decreasing order of contaminant formation indicated above, that is, frying > grilling > oven-broiling, was confirmed in this study, since the highest formation of acrylamide was obtained after frying (129 µg kg^−1^) and the lowest after oven-broiling (31.3 µg kg^−1^). The acrylamide formation after grilling was intermediate, in the range of 30.7–72.5 µg kg^−1^.

### 2.5. Overall Remarks

It is well-known that red meat has been classified as probably carcinogenic to humans (by IARC), and this food safety aspect relies on inherent characteristics of food matrices, such as the presence of heme iron mediating lipid peroxidation, neuraminic acid, etc. [57,58,59], but also on heat processing, which leads to the formation of several toxic compounds. Based on the consulted literature about the role of meat heating processes on the production of contaminants, such as HAAs, NAs, PAHs and acrylamide, the effect of meat heating processes on the number and level of these molecules was highlighted. Many researchers, indeed, report that the increase of carcinogenic HAAs, NAs and PAHs is substantially higher when meat is cooked by frying if compared to grilling and oven-broiling [34,44,48]. This finding can be due to particular steps in vegetable oils’ (i.e., sunflower, corn, soybean, coconut oil, etc.) production, such as drying before the oil extraction, which can cause the formation of PAHs [60]. The oil type also influences the increase in contaminants after heating processes, as in the case of HAAs, the formation of which is directly proportional to the level of saturated fatty acids [61]. Quantifiable levels of some NAs, namely, N-Nitrosodiethylamine, N-Nitrosodimethylamine, N-Nitrosomethylethylamine, N-Nitrosopiperidine and N-Nitrosomorpholine, in the range 0.1–0.7 µg kg^−1^, were also reported by different authors in several oil types, such as olive oil, grapeseed, sunflower, perilla, sesame, rape, soybean, corn and peanut [23,24,25,26,30,45,46,47,48,49,50,51,52,53,54,55,56,57,58,59,60,61,62]. Regarding acrylamide, most studies were focused on meat heating processes by frying, confirming the possible formation of such a contaminant up to 129 µg kg^−1^. This last finding is worthy of attention since such concentrations are higher than the benchmark dose established for wheat-based bread and soft bread other than wheat-based bread (equal to 50 and 100 µg kg^−1^, respectively) in the European Commission Regulation (EU) 2017/2158 (European Commission 2017, to be revised in 2023) [63], which establishes mitigation measures and benchmark levels for the reduction of the presence of acrylamide in food.

After establishing the prominent role of frying in the formation of carcinogenic substances in cooked meat, it is also possible to affirm that other heating processes influence the formation of these substances in meat depending on the time and temperature of treatment. In this regard, oven-broiling seems to be safer than grilling. This can be due mainly to the possible direct contact with fire, and then the higher temperatures reached, since the heating time used in oven-broiling is usually higher than grilling. Finally, the few findings related to meat cooked by boiling, microwaving and sous vide report that these types of heating treatment lead to the lowest formation of both HAAs, NAs and PAHs in beef and pork [61,62,64] (Table 1).

Gibis and Weiss [16] compared the increase of HAAs in different meat species after grilling, verifying that the highest effect was obtained in beef (total HAAs = 7.74 µg kg^−1^), followed by pork (5.54 µg kg^−1^), lamb (4.97 µg kg^−1^) and horse meat (4.78 µg kg^−1^). The production of 4,8-DiMeIQx, considered the highest carcinogenic compound, was detected in roasted beef and in pork meat, showing comparable concentrations: 0.9 µg kg^−1^ [19] and 0.7 µg kg^−1^ [34], while Xiao et al. [65] did not detected quantifiable amounts of this compound in roasted lamb meat. Skog et al. [31] investigated HAA production in cattle, pork and lamb meat fried under the same conditions in terms of temperature and time of process. The results, expressed as the sum of MelQx, PhIp and 4,8-DiMeIQx, demonstrated that the highest increase was found in lamb meat, followed by pork and beef: 2.4, 1.5, 1.0 µg kg^−1^.

Regarding NAs, it is possible to compare data related to grilling and frying treatments. Taking into account the sum of N-Nitrosodimethylamine, N-Nitrosopiperidine and N-Nitrosopyrrolidine detected after grilling at 200–250 °C at a distance of 3–10 cm from the charcoal, the highest increase (31.9 µg kg^−1^) was reported in beef meat by Mirzazadeh et al. [22], followed by that reported in pork meat (11.3 µg kg^−1^) by Yurchenko and Molder [23] and the amount quantified by Kocak et al. [32] in lamb meat cooked by charcoal for 16 min (2.7 µg kg^−1^). The comparison after frying substantially confirms this trend, since the highest amount of these three NAs (15.6 µg kg^−1^) was reported by Mirzazadeh et al. [22] in beef meat, followed by those reported by Yurchenko and Molder [23] in pork and mutton meat (12.4 and 4.6 µg kg^−1^, respectively). It is worth noting that NA analytical determination is heavily influenced by experimental conditions (e.g., time and temperature) and the analytical technique adopted. Thus, these deductions need to be confirmed through ad hoc studies. Olatunji et al. [27] reported different PAH levels (as the sum of four PAHs, Benzo[k]fluoranthene, Benzo[a]pyrene, Indeno[123-cd]pyrene and Benzo[ghi]perylene) in grilled beef (6.7 µg kg^−1^) and pork (9.0 µg kg^−1^), while no significant difference was registered after boiling for these two types of meat (beef: 2.4 µg kg^−1^, pork: 2.2 µg kg^−1^). The amount of these four PAHs in charcoal-grilled mutton, detected by Eldaly et al. [33], was significantly lower, equal to 0.7 µg kg^−1^. Eldaly et al. [33] also published data about the sum of 16 PAHs after frying beef and mutton meat. The total amount detected in mutton meat was higher than that quantified in beef meat, corresponding to 4.6 and 1.1 µg kg^−1^, respectively. Lastly, the formation of HAAs and NAs seems to be higher in beef if compared to pork and lamb meat; meanwhile, the same comment cannot be made relating to PAH formation.

In light of these considerations, the possible inhibition of toxic compound formation in meat could be a very interesting future perspective to apply to all heating processes of meat from different animal species at both domestic and industrial scales. Previous research proposed the use of antioxidants to reduce or inhibit the formation of carcinogenic substances, showing encouraging results (e.g., [66,67,68,69,70]); particularly, an overview of the promising strategy to mitigate the risk of HAAs by natural compounds has been reported by Nadeem et al. [14]. In Table 2, the ingredients, mainly plant extracts, tested as possible inhibitory agents of different toxic compounds in meat are summarized.

## 3. Effect of Heating Processes on Meat Nutritional Value

Generally, the heating process contributes to the loss of water-holding capacity, resulting in the different concentrations of proteins, fat and ash compared to raw meat [71,72]. Table 3 summarizes the effect of the heating process on the nutritional value (fatty acids, proteins, vitamins, and minerals) of meat from different animal species described in the following sections.

### 3.1. Lipids

An increase of total lipids, SFA and MUFA and a change of PUFA content was observed in meat from cattle, horse, lamb and pork compared to raw meat, which occurs with different rates according to heating processes.

In beef from Alentejano bulls, Alfaia et al. [73] reported an increase in total lipids from 1.25 to 2.21%, 2.17% and 2.61% in grilled, boiled and microwaved beef, respectively; meanwhile, all heating processes showed an increase of SFA and MUFA content compared to raw meat (from 39.3 g/100 g to about 43 g/100 g and from 29.11 g/100 g to about 32 g/100 g, respectively).

Furthermore, the authors observed a decrease in PUFA from 24.69 to 17.60 g/100 g, 18.77 g/100 g and 18.84 g/100 g, in grilled, boiled and microwaved meat, respectively, while, CLA revealed a greater stability to the thermal process. Significant changes in total lipids content were also observed in meat from unweaned Limousin calves with an increase from 0.7 to 1.01, 0.89 and 1.37% in grilled, sous vide and steamed meat, respectively, compared to raw meat [75]. Moreover, the authors found that heat treatments increased lipid oxidation (TBARS value) as follows: steaming > sous vide > grilling.

In foal meat, Dominguez et al. [88] observed a decrease in SFA (from 33.12 g/100 g to 30.95, 31.89, 30.03, 20.18 g/100 g) and an increase in MUFA (from 20.93 g/100 g to 21.78, 21.80, 21.95, 52.53 g/100 g of fat) in raw, grilled, roasted, microwaved and fried meat, respectively. On the contrary, PUFA content was affected only by the frying method, which led to a decrease from 37.64 to 19.21 g/100 g. Meanwhile, the other heating processes did not impact this group of fatty acids.

The temperature and degree of doneness tend to have considerable influence on the lipid content and on the fatty acids profile because it can be influenced by the reduction of moisture as well as the use of a lipid source in the heating processes. Frying is the heating process that causes the greatest impact with the highest increase of total lipid and MUFA due to the incorporation of lipids from the oil used during heating processes [4,5,6,7,8,9,10,11,12,13,14,15]. Recently, Borela et al. [89] studied the influence of different frying methods on the physiochemical characteristics of fillet steaks, highlighting that an air fryer can be considered a healthy method to cook meat. Particularly, the authors found that the total lipids content increased from 3.1 g/100 g to 3.58, 3.65 and 9.51 g/100 g in pan-fried without oil, air-fried and pan-fried with oil meat, respectively, compared to raw meat.

An increase in total fatty acid composition as affected by heating processes was observed in lamb meat [82], with SFA increasing from 1.039 mg/g to 1.245 and 1.468 mg/g, MUFA increasing from 1.125 mg/g to 1.355 and 1.557 mg/g and PUFA increasing from 187 to 237 and 280 mg/100 g in raw, sous vide and grilled meat, respectively. Particularly, these authors found that the extent of the increase in the concentration of individual PUFA fatty acids was greater in lambs fed at pasture than those fed cereal base concentrates.

### 3.2. Proteins

Heating processes affect the physiochemical state of proteins and the bioavailability of their amino acids, as suggested by Lopes et al. [77]. These authors found that grilling was the treatment that changed the amino acid composition the most, showing the highest content of leucine (23 g/100 vs. 11.7, 11.9, 12 g/100 g detected in raw, microwaved and boiled meat, respectively) and the lowest values of glutamic acid, threonine and serine. It has been established that the effects of cooking temperatures on meat proteins are varied [90]. Particularly, the heating temperature was a determining factor with respect to the stability of aromatic amino acid residues. Gatellier and Sante-Lhoutellier [91] found that heating bovine meat in a dry bath for 30 min at 60 °C did not affect the stability of amino acids, but the contents of these amino acids were highly reduced when cooked at 100 and 140 °C, decreasing as follows: tryptophan > phenylalanine > tyrosine. A decrease of tryptophan during heating processes was observed also in pork meat, with a 50% reduction of the content of these amino acids when cooked in the oven at 102 °C for 20 min [85]. Boiling pork meat at 60 and 75 °C decreased the protein content by 11% and 18%, respectively, compared to raw meat; histidine was the most affected amino acid, showing reductions of 17% and 31%, respectively [84]. Olagunju and Nwachukwu [83] found no differences in total and in essential amino acids of goat meat grilled or boiled compared to raw meat. Meanwhile, these authors found that heating significantly affected the amino acid content of cow meat with a decrease of tyrosine > tryptophan > glycine in meat boiled or grilled compared to raw meat.

Bhat et al. [15] highlighted that heating processes affect the digestibility of meat proteins in the gastrointestinal tract. This is prominent from a nutritional point of view because the nutritional value of muscle proteins is affected by protein digestibility and by the presence of end products such as amino acids or peptides that are present in an assimilable form for absorption. Several studies reported that the functional value of meat, in terms of bioactive compound contents, can also be altered during heating processes. Particularly, a decrease in creatine (from 384.9 to 311 mg/100 g), taurine (from 49.6 to 24.3 mg/100 g), carnosine (from 409.6 to 323.5 mg/100 g) and CoQ10 (from 1.55 to 1.3 mg/100 g) concentrations was found in lamb meat grilled at 70 °C compared to raw meat [92]. Glutathione concentrations also showed a decrease (−41%) in beef after the grilling process (from 14.87 to 12.46 mg/100 g), as reported by Rakowska et al. [93].

The effects of household heating process preparations on bioactive properties (antihypertensive activity and antioxidative capacity) of the semimembranosus muscle from pork and beef were studied by Jensen et al. [72]. These authors reported that heating processes in a heated pan (75 °C internal T for 20 min) did not affect the retention of lipids or the sum of amino acids, but reduced the amount of the free amino acid taurine and affected the antioxidative capacity negatively.

### 3.3. Minerals and Vitamins

Meat contains several bioavailable micronutrients, many of which are water-soluble so the heating process and temperature could be a determinant for the magnitude of mineral loss.

A decrease in the mineral content from 13.6% to 21.1% was observed in deep-fried, pan-fried, oven-cooked and microwaved beef steak, with microwaving causing the highest loss [80]. The microwave heating process also determines a decrease in Ca, Cu, Fe, K and Mg contents of bovine liver [81]; meanwhile, these authors reported that the mineral content was stable in bovine liver cooked with the sous vide method.

Grilling and boiling cooked methods decrease the contents of Na, K, P, Ca and Mg, while increasing the Fe and Zn contents in pork and cattle meat [79].

Purchas et al. [94] compared the mineral content in uncooked and cooked lean beef, highlighting a decrease in the contents of Na and K and an increase in Ca, Cu, Fe, Mn and Zn in cooked meat compared with raw meat. An increase in the total mineral component, expressed as ash content (from 3.76 to 13.84%), was found in grilled and boiled sheep meat (Olagunju et al. [83]). These results indicate that the same minerals (e.g., divalent minerals) are better retained during heating processes, which may be due to their greater association with protein, as suggested by Sobral et al. [4].

A different impact of heating processes has been reported on vitamin content due to the different heat stabilities of the water-soluble (B-complex and C) and the fat-soluble (A, D, E, and K) vitamins. Kaliniak-Dziura et al. [75], in meat from unweaned Limousin calves, found that heat treatments decreased the level of Vitamin E from 163.88 µg/100 g to 140.88, 124.06 and 107.69 µg/100 g in steamed, sous vide and grilled meat, respectively.

Vitamin B12 is relatively resistant to higher temperatures. In beef, Czerwonka et al. [78] found that roasting and grilling have little effect on the content of vitamin B12, while frying reduces the content of this vitamin by 32% compared with raw meat.

On the contrary, Vitamin B1 is the most heat-labile vitamin of the B-complex vitamins. As reviewed by Sobral et al. [4], several different heating processes reduce the content of vitamin B1. Gerber et al. [79] reported that 75% of vitamin B1 is lost in grilled pork and 100% is lost in boiled beef brisket.

### 3.4. Overall Remarks

Heating processes impact meat’s nutritional value, affecting the intake of essential nutrients. Heat treatment leads to a change in lipids and the fatty acid composition of meat due to mechanisms such as water loss, lipid oxidation and diffusion exchange, which occur during heating processes [95]. Frying is the heating process that induces the greatest increase of meat lipid content, more than eight times for MUFA, and a strong decrease in PUFA content. [96] The immersion of meat in oil is a crucial heating process parameter affecting the fat content and the fatty acids profile as a result both of water loss due to the high temperature and an exchange of fatty acids between the frying oil and the animal tissue.

Heating processes under relatively mild conditions (T < 100 °C) showed an increase in PUFA content [95]. Therefore, grilled and sous vide heating processes would be adequate heating process techniques to preserve a beneficial fatty acid profile in meat with a greater n3 PUFA content than other heating processes, as observed in horse and lamb meat [82,88]. Particularly, in the sous vide samples, the oxidation process was limited due to the evacuation of oxygen during the vacuum-packing process, preventing PUFA from oxidation, as suggested by Modzelewska-Kapituła et al. [96].

It has been observed [15] that the sous-vide heating process also induces favorable changes in muscle proteins (e.g., partial unfolding or exposure of cleavage site), improving their digestibility compared to other heating processes, such as stewing and roasting, which induce unfavorable changes (e.g., protein aggregation, severe oxidation), decreasing protein digestibility. In addition, sous vide heating processes improve the bioavailability of mineral elements such as Cu, Fe and Zn compared to raw meat, as highlighted by da Silva et al. [81]. These observations are in agreement with Suleman et al. [97] who highlighted how modern heating technologies (e.g., microwave, infra-red radiation, sous vide heating processes) can help to improve the quality of lamb meat. In particular, these methods lead to a better retention of nutrients, no oxidative reaction and greater stability of heat-sensitive vitamins and nutrients.

Although heating processes impact nutritional properties, other factors such as animal species could contribute to mitigating its effects. In particular, horse meat shows a favorable fatty acid profile, with the highest PUFA content compared to beef because the horse, being monogastric, can efficiently digest and absorb the lipids introduced with the diet without rumen biohydrogenation taking place. In addition, horses, as well as sheep, are usually reared at pasture so the natural antioxidants from pasture could protect polyunsaturated fatty acids in horse and lamb meat during the heating treatments. Previous studies [88,98] reported that cooked meat from pasture-fed animals presented higher oxidative stability than meat from animals fed with a concentrated-based diet.

## 4. Concluding Remarks and Future Perspectives

Heating processes differently impact toxic compound production and the nutritional value of meat from different animal species. On the basis of the large number of compounds discussed in this review, it is possible to extract interesting and significant advice, which is very useful for a meaningful risk–benefit evaluation on this topic.

The meat heating process of grilling was the most studied method in the literature, and this type of heating process leads to the possible increase of many toxic compounds; meanwhile, the effect on nutritional value depends on the animal species.

Boiling represents a good meat heating process, since the literature only reports a slight increase of PAHs and HAAs, while a decrease in PUFA and vitamins and minerals was observed in beef. The literature data on microwaving and sous vide, which are innovative heating processes, highlighted that, apart from the slight formation of HAAs, these two procedures do not lead to significant formation of toxic compounds, showing, in addition, an improvement in meat nutritional quality.

It is worth noting that, looking at the overall increase of different classes of toxic compounds, it is not possible to establish a correlation to the animal species; meanwhile, the changes in the nutritional properties of meat vary related to the different animal species. Particularly, the increase or the decrease in saturated, monounsaturated and polyunsaturated fatty acids depends on animal species, which could contribute to mitigating the effect of heating processes as discussed above. In addition, heating processes have an important role in improving the protein digestibility of red meat from different animal species. It is possible to suggest that heat treatments of red meat under relatively mild conditions (T < 100 °C) are useful to preserve some nutritional parameters and avoid significant formation of toxic compounds at the same time.

As a future perspective, in order to evaluate the risk–benefit related to red meat consumption after the heating process, planning an ad hoc experimental design to study the modification both on nutritional and safety meat quality is required. This approach, supported by statistics/chemometrics tools, is useful to assess the effects of heating treatments on red meat’s overall quality. The obtained findings could be exploited to develop novel meat product formulations with increased safety and nutritional quality, given the increasing consumer concerns and requests about this topic.

## Figures and Tables

**Figure 1 foods-13-00445-f001:**
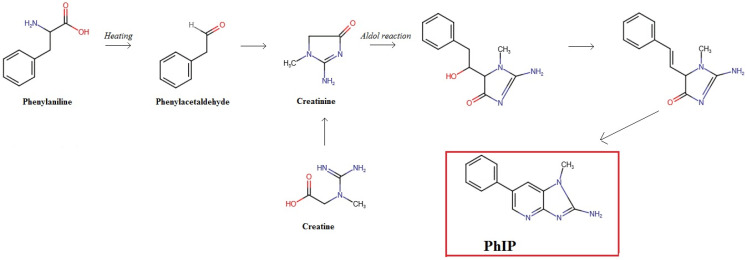
Mechanism of formation of PhIP from phenylalanine and creatine (adapted by Gibis, 2016 [39]).

**Figure 2 foods-13-00445-f002:**
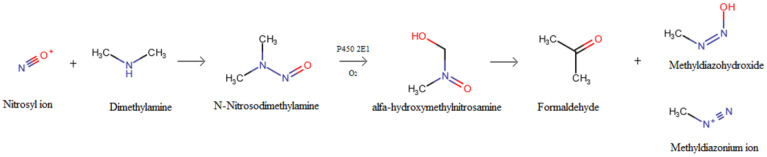
Mechanism of formation of N-Nitrosodimethylamine and its activated products [44].

**Figure 3 foods-13-00445-f003:**
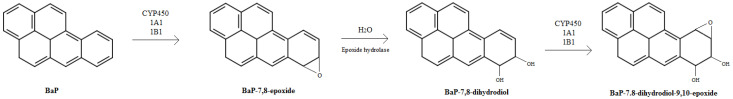
Mechanism of activation of benzo-(a)-pyrene (adapted by Moserová et al., 2009 [50]).

**Figure 4 foods-13-00445-f004:**
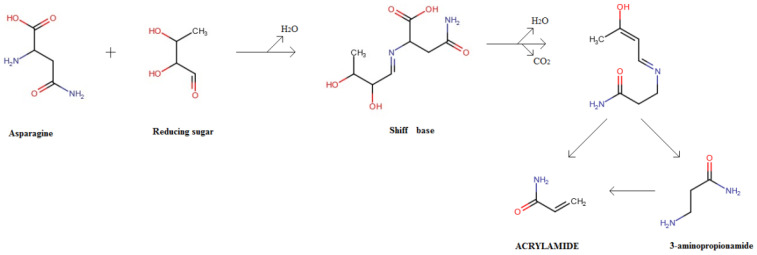
Typical mechanism of formation of acrylamide in food (adapted by Augustine and Bent, 2019 [55]).

**Table 1 foods-13-00445-t001:** Effect of heating processes on the production of processing contaminants in red meat from different animal species.

	Heating Processes					
	Grilled	Roasted/Oven-Broiled	Boiled	Microwaved	Fried	Sous Vide
Cattle	Heterocyclic amines increase in meat [16,17]	Heterocyclic amines increase [18,19]			Heterocyclic amines increase in meat [20]	Heterocyclic amines increase [21]
	Nitrosamines increase in beef preparations [22] and meat [23]			No increase of nitrosamines in beef preparations [22]	Nitrosamines increase in beef preparations [22]	
	Polycyclic aromatic hydrocarbon increase [24,25,26,27,28]	Polycyclic aromatic hydrocarbon increase [29]	Polycyclic aromatic hydrocarbon increase [27]		Acrylamide increase in burger [30]	
Sheep and Goat	Heterocyclic amines increase in meat [16]				Heterocyclic amines increase in lamb meat [31]	
	Nitrosamines increase in meat [32]				Nitrosamines increase in meat [23]	
	Polycyclic aromatic hydrocarbon increase [25,33]					
Pig	Heterocyclic amines increase in meat [16,34]	Heterocyclic amines increase in meat [34]	Heterocyclic amines (MeIQx) increase in meat [34]	Heterocyclic amines increase in bacon [35]	Heterocyclic amines increase in bacon [35]	
	Nitrosamines increase in meat [23]				Nitrosamines increase in meat [23]	
	Polycyclic aromatic hydrocarbon increase [27]	Polycyclic aromatic hydrocarbon increase [27,36]	Polycyclic aromatic hydrocarbon increase [27]			
Horse	Heterocyclic amines increase in meat [16]					

**Table 2 foods-13-00445-t002:** Ingredients tested as possible inhibitory agents of toxic compounds formation in meats.

Type of Contaminant	Ingredients Added in Meat Preparations	References
HAAs	Coriander seeds, cumin, finger root, galangal, rosemary, pepper, garlic, garlic essential oil, onion, turmeric, oregano, paprika, curry leaf, cinnamon and clove, lemon grass, fennel, prickly ash peel and star anise	[36,66]
PAHs	Garlic, onion, nutmeg oil nanoemulsion, ginger powder, green tea marinade, sesamol	[21,36,66,67]
NAs	Garlic, strawberry and garlic juice, garlic powder, onion, green tea, grape seed extract	[29]

**Table 3 foods-13-00445-t003:** Effect of heating processes on nutritive value of red meat from different animal species.

Heating Processes						
	Grilled	Roasted/Oven Broiled	Boiled	Microwaved	Fried	Sous vide
Cattle	Total lipids, SFA and MUFA increase, PUFA decrease [73]		Total lipids, SFA and MUFA increase, PUFA decrease [73]	Total lipids, SFA and MUFA increase, PUFA decrease [73]	Total lipid increase, SFA increase [74]	Total lipids increase in Limousin calves [75]
	Protein digestibility decrease [76] Leucine, glutamic and aspartic acid increase [77]	Protein digestibility increase [76]		Leucine, glutamic and aspartic acid increase [77]	Protein digestibility increase [76]	
	Vitamin E decrease [75] No effect on vitamin B12 [78]	No effect on vitamin B12 [78]	Vitamin B1 and vitamin D decrease [79]		Vitamin D decrease; vitamin B12 decrease [78]	Vitamin E decreases [75]
	Ash increase from 1 to 1.3% [72] Na, K, P, Ca and Mg decrease Fe and Zn increase [79]		Na, K, P, Ca and Mg decrease, while Fe and Zn increase [79]	Fe, Mg and K decrease [80] Ca, Cu, Fe, K, Mg decrease in bovine liver [81]		Ca, Cu, Fe, K, Mg stability in bovine liver [81]
Sheep and Goat	SFA, MUFA and PUFA increase [82]		Ash increase [83]			SFA, MUFA and PUFA increase [82]
	Ash increase [83] No differences in total amino acids and in essential amino acids [83]		No differences in total amino acids and in essential amino acids [83]			
Pig	Total lipids and ash increase [81]		Total amino acids content decrease [84] Tryptophan decrease [85]			Protein digestibility increase [86]
	Vitamin B1 strong decrease [79]		Protein digestibility increase [87]			
	Ash increase [72]; Na, K, P, Ca and Mg decrease, while Fe and Zn increase [79]		Na, K, P, Ca and Mg decrease, while Fe and Zn increase [79]			
Horse	Total lipids, MUFA and PUFA content increase; SFA content decrease [88]	Total lipids, MUFA and PUFA content increase; SFA content decrease [88]		Total lipids, MUFA and PUFA content increase; SFA content decrease [88]	Total lipids, MUFA and PUFA content increase; SFA content decrease [88]	

## Data Availability

No new data were created or analyzed in this study. Data sharing is not applicable to this article.

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
