# Peer review of "Red Meat Heating Processes, Toxic Compounds Production and Nutritional Parameters Changes: What about Risk–Benefit?"

_foods, 2024, doi:10.3390/foods13030445_

Round 1

Reviewer 1 Report

Comments and Suggestions for Authors

The review is comprehensive and covers essential aspects of toxic compound generation during the cooking process of red meat. I have some suggestions for improvement:

1. In addition to focusing on the neo-formed compounds generated during the cooking of red meat, I would suggest the authors to add other axes regarding the "toxicity" of red meat based on other mechanistic hypotheses (ref: https://iopscience.iop.org/article/10.1088/1755-1315/426/1/012176/meta). These mechanisms include the formation of toxic aldehydes derived from heme iron-mediated lipid peroxidation (the works of Fabrice Pierre) and neuraminic acid (the works of Annie N Samraj). I believe that adding these mechanistic hypotheses would significantly improve the manuscript.

2. I would suggest the authors to develop a deeper analysis on how to render red meat "safer to eat" through different strategies based on the available scientific evidence. In particular, some cooking recommendations would be highly appreciated.

Thank you.

Author Response

The authors would like to thank the Academic Editor and the reviewers for their effort in improving the scientific impact of the Paper. The manuscript has been revised, according to suggestions and comments, editing corrections and rewording the text where necessary. Please note that references to lines are referred to pdf.

Reviewer 1 comments:

The review is comprehensive and covers essential aspects of toxic compound generation during the cooking process of red meat. I have some suggestions for improvement:

  1. In addition to focusing on the neo-formed compounds generated during the cooking of red meat, I would suggest the authors to add other axes regarding the "toxicity" of red meat based on other mechanistic hypotheses (ref: https://iopscience.iop.org/article/10.1088/1755-1315/426/1/012176/meta). These mechanisms include the formation of toxic aldehydes derived from heme iron-mediated lipid peroxidation (the works of Fabrice Pierre) and neuraminic acid (the works of Annie N Samraj). I believe that adding these mechanistic hypotheses would significantly improve the manuscript.

Answer: Thanks for your suggestion. This review was mainly focused on the role of heat treatments on toxic compounds formation and nutritional characteristic modifications in red meat. Thus, some inherent properties of red meat linked to toxic effects, such us heme-iron presence, were not taken into account. However, following the referee’s suggestion, a new sentence, together with 3 new references has been added at lines 263-267.

 I would suggest the authors to develop a deeper analysis on how to render red meat "safer to eat" through different strategies based on the available scientific evidence. In particular, some cooking recommendations would be highly appreciated.

Answer: Thanks for your comment. Last section of “Concluding remarks” and table 3 were included as a final suggestion with regard to “safer to eat” red meat. Following the referee’s suggestion, new comments have been added at lines 547-550.

Reviewer 2 Report

Comments and Suggestions for Authors

This suggestion is due to some toxic effects on humans, such as cardiovascular disease and cancer, there is a risk that these molecules may become active, especially when cooked. - the term "Cooking" was used throughout the publication, which in my opinion is not the right wording. Cooking is the thermal processing of meat at the boiling point of water (100 C), which has no relation to the presented content. Toxic and potentially toxic compounds may be formed at temperatures much higher than 100 C. This is particularly problematic when the authors blame "cooking" for the potential toxicity of processed meat. As for harmful compounds, few examples are given. Moreover, many of them are not recent reports (not older than 5 years). When it comes to nutritional compounds, this part does not provide new information, but rather presents known relationships. Therefore, in my opinion, the study is more of a report than a review of the latest data. For this reason, it does not meet the novelty criterion and should not be published in Foods.

Author Response

The authors would like to thank the Academic Editor and the reviewers for their effort in improving the scientific impact of the Paper. The manuscript has been revised, according to suggestions and comments, editing corrections and rewording the text where necessary. Please note that references to lines are referred to pdf.

Reviewer 2 comments:

This suggestion is due to some toxic effects on humans, such as cardiovascular disease and cancer, there is a risk that these molecules may become active, especially when cooked. - the term "Cooking" was used throughout the publication, which in my opinion is not the right wording. Cooking is the thermal processing of meat at the boiling point of water (100 C), which has no relation to the presented content. Toxic and potentially toxic compounds may be formed at temperatures much higher than 100 C. This is particularly problematic when the authors blame "cooking" for the potential toxicity of processed meat. As for harmful compounds, few examples are given. Moreover, many of them are not recent reports (not older than 5 years). When it comes to nutritional compounds, this part does not provide new information, but rather presents known relationships. Therefore, in my opinion, the study is more of a report than a review of the latest data. For this reason, it does not meet the novelty criterion and should not be published in Foods.

Answer: We agree with reviewer that cooking is a process which applies temperatures up to 100 °C. Thus, the expression “cooking or boiling” has been replaced by “heating processes” both in the title and in the whole manuscript. The manuscript has also been improved providing new information based on recent references. Overall, 13 new references have been added.

Reviewer 3 Report

Comments and Suggestions for Authors

In the article title “Red Meat Cooking, Toxic Compounds Production and Nutritional Parameters Changes: What About Risk-Benefit?, the authors provides an updated overview on the effects of different cooking methods on toxic compounds production and nutritional parameters of red meat. The specific comments are listed below.

1.The novelty and necessity need to be highlighted. The Introduction part lacks innovation. The entry point of the manuscript is good, but there are already quite a few publications and reviews on toxic compounds production and nutritional parameters of meat or meat products.

2.Page 3: Table 1. The formation of polycyclic aromatic hydrocarbons is closely related to the environment. The wood used for smoking or grilling can promote the formation of polycyclic aromatic hydrocarbons. It is recommended to indicate the specific grilling material, whether charcoal grilling or other grilling materials are used.

3.Page 8, Line 254-264: It is not necessary to take up a lot of space to simply list the relevant studies on the toxic compounds of meats, these studies can be summarized and condensed in the form of a table, which is more direct and clear.

4.Page 8, Line 265-273: Not only nitrosamines, but also other harmful substances. Processing conditions have a significant impact on the formation of harmful substances in meat, so it is difficult to compare without knowing specific conditions, as the content may also be related to the varieties, parts, pre-treatment methods, and other factors of the meat. Additionally, the measurement method can also affect the level of detection.

5. It is recommended to consider article segmentation again, an example of a paragraph, really like a list, the lack of logical correlation.

6.At present, there are no specific regulations on the limit values for most toxic compounds. After reading so much literature and summarizing, can you provide any suggestions or opinions?

7.Page 14, Line 451-453: This part of the conclusion is somewhat unreasonable, the existing literature has shown that the increase of toxic compounds is related to animal species.

8.Whether the authors considered the toxic compounds and corresponding nutritional properties of meat in the same study is more convincing, because the conditions of meat products treated by different cooking methods are not consistent, and it is difficult to establish a link between different studies.

9. The entire manuscript feels more like a simple listing of existing research results without the author's in-depth summaries. There is still a lot of room for improvement in writing.

Comments on the Quality of English Language

Minor editing of English language required.

Author Response

The authors would like to thank the Academic Editor and the reviewers for their effort in improving the scientific impact of the Paper. The manuscript has been revised, according to suggestions and comments, editing corrections and rewording the text where necessary. Please note that references to lines are referred to pdf.

Reviewer 3 comments:

In the article title “Red Meat Cooking, Toxic Compounds Production and Nutritional Parameters Changes: What About Risk-Benefit?”, the authors provides an updated overview on the effects of different cooking methods on toxic compounds production and nutritional parameters of red meat. The specific comments are listed below.

Answer: Thanks for your remark. The novelty of this review has been highlighted and further references on toxic compounds production and nutritional parameters of meat have been added (lines 68-83).

2.Page 3: Table 1. The formation of polycyclic aromatic hydrocarbons is closely related to the environment. The wood used for smoking or grilling can promote the formation of polycyclic aromatic hydrocarbons. It is recommended to indicate the specific grilling material, whether charcoal grilling or other grilling materials are used.

Answer: Thanks for your remark. Ten references listed in table 1 have been carefully checked and all useful info has been added in the text at lines 111, 123, 125, 216-217.

3.Page 8, Line 254-264: It is not necessary to take up a lot of space to simply list the relevant studies on the toxic compounds of meats, these studies can be summarized and condensed in the form of a table, which is more direct and clear.

Answer: Thanks for your comment. The authors would prefer not using another table which, in practice, would overlap to table 1. Few data are listed in this section, so that they can be easily commented in a paragraph without compromising readability. We hope the referee can agree with this answer.

4.Page 8, Line 265-273: Not only nitrosamines, but also other harmful substances. Processing conditions have a significant impact on the formation of harmful substances in meat, so it is difficult to compare without knowing specific conditions, as the content may also be related to the varieties, parts, pre-treatment methods, and other factors of the meat. Additionally, the measurement method can also affect the level of detection.

Answer: The “Overall remarks” paragraph has been designed and organized so to resume the results obtained under the same conditions obtaining reliable comparisons. The section at lines 310-322 has been focused on N-Nitrosamines and this is the reason why only this class of contaminants has been cited. Following the reviewer’s suggestion, another comment has been added at lines 319-322.

  1. It is recommended to consider article segmentation again, an example of a paragraph, really like a list, the lack of logical correlation.

Answer: Thanks for your comment. The review has been organized in 2 sections, the first (Section 2, par. 2.1 – 2.4) focused on “safety” aspects, the second (Section 3, par. 3.1 – 3.3) focused on “nutritional” aspects. Both sections have been “closed” by drafting an “Overall remarks” paragraph (2.5 and 3.4). Both sections have also been simplified by adding tables 1-2 which select most significant studies, so to simplify the reading of the whole review. We believe that this review organization can be considered as effective and proper for this type of review article.

6.At present, there are no specific regulations on the limit values for most toxic compounds. After reading so much literature and summarizing, can you provide any suggestions or opinions?

Answer: According to the referee’s suggestion, new comments and a new reference have been added at lines 171-185.

7.Page 14, Line 451-453: This part of the conclusion is somewhat unreasonable, the existing literature has shown that the increase of toxic compounds is related to animal species.

Answer: Thanks for your remark. The sentence has been modified, since the comment was referred to the overall evaluation of different toxic compounds, not to a specific one (lines 540-543). Indeed, the “view” of this review article was not the evaluation of a single class of toxic compound but the overall evaluation of all classes. Moreover, as highlighted at lines 50-58, the evaluation of so-called “cocktail effect” represents a new frontier in food safety knowledge. No study is available regarding synergistic toxic effect possibly caused by HAAs, NAs, PAHs and acrylamide due to the consumption of cooked red meat. Thus, this review also aims at encouraging such type of studies and further research.

8.Whether the authors considered the toxic compounds and corresponding nutritional properties of meat in the same study is more convincing, because the conditions of meat products treated by different cooking methods are not consistent, and it is difficult to establish a link between different studies.

Answer: The authors agree with the referee. Indeed, in different sections, the review remarks the lack of studies focused on the simultaneous evaluation of different classes of toxic compounds in red meat after heat treatments and the need of such studies. In this regard, the underlining of such lack in knowledge is an aim of this this review which provides an overview of the available knowledge together with a comprehensive discussion. In order to stress more this aspect, another comment has been added at lines 551-556.

  1. The entire manuscript feels more like a simple listing of existing research results without the author's in-depth summaries. There is still a lot of room for improvement in writing.

Answer: Following the reviewer’s suggestion, other sections devoted to concluding remarks, overall evaluations and 13 new references have been added (263-267, 319-322, 335-342, 547-556).

Reviewer 4 Report

Comments and Suggestions for Authors

This review is well structured and covers the main cooking methods for red meat. However, there are some areas that need to be revised:

1. Some studies/articles are not cited in the references (e.g. EFSA - line 50).

2. Phrases and opinions that need to be substantiated, as some even mention that studies have been consulted (lines 34, 45, 62, 227,...).

3. Sources of figures (Fig. 2)

Also in the conclusion, the use of a table with information that should be in the description of the process is incomprehensible. Suggestion to reformulate.

Finally, an important point - the first question is not well explained and leaves many doubts. It is suggested that more attention be paid to this (topical) issue and that the conclusions be more objective, in line with the question in the title.

Author Response

The authors would like to thank the Academic Editor and the reviewers for their effort in improving the scientific impact of the Paper. The manuscript has been revised, according to suggestions and comments, editing corrections and rewording the text where necessary. Please note that references to lines are referred to pdf.

Reviewer 4 comments:

This review is well structured and covers the main cooking methods for red meat. However, there are some areas that need to be revised:

  1. Some studies/articles are not cited in the references (e.g. EFSA - line 50).

Answer: Thanks for your remark. All references have been checked for confirming their citation in the main text. The reference EFSA 2023 has been added at line 183.

  1. Phrases and opinions that need to be substantiated, as some even mention that studies have been consulted (lines 34, 45, 62, 227,...).

Answer: Following the reviewer’s suggestion, new references together with other comments have been added at lines 38, 47, 64, 272, 297.

  1. Sources of figures (Fig. 2)

Answer: Thanks for your in-depth work of revision. A reference to Figure 2 has been added at line 148-150.

Also in the conclusion, the use of a table with information that should be in the description of the process is incomprehensible. Suggestion to reformulate.

Answer: According to the referee’s suggestion, table 3 was moved to the end of section devoted to contaminants (now Table 2, line 349). Indeed, this section has only been added to give the reader a brief “overview” of the possible strategies proposed to inhibit toxic compounds formation in red meat after cooking. This is not a conclusion but a specific section the author consider as very significant for this topic in the view of future research. Thus, in order to avoid any misunderstanding, this section was moved.

Finally, an important point - the first question is not well explained and leaves many doubts. It is suggested that more attention be paid to this (topical) issue and that the conclusions be more objective, in line with the question in the title.

Answer: Thanks for your remark. Following the reviewer’s suggestion, the last paragraph “Concluding remarks” (now “Concluding remarks and future perspectives”) has been improved, so to give the reader a final overview of available knowledge about risk-benefit of red meat consumption after heat processing. Final recommendations for future studies have been added as well (lines 551-556).

Round 2

Reviewer 2 Report

Comments and Suggestions for Authors

-

Author Response

Dear Editors,

We would like to thank you for the timely work carried out in the management of the submitted paper. Please find enclosed the revised version of the manuscript foods-2785866. The last comment by Reviewer n.4 has been considered and the manuscript has been revised accordingly.

I hope you will find this review suitable for publication in the Foods Special Issue entitled: “Strategies to Improve the Functional Value of Meat and Meat Products”.

Sincerely yours,

Dr. Marco Iammarino on behalf of co-authors

Reviewer 4 Report

Comments and Suggestions for Authors

The article has been restructured to give it more scientific consistency and to correct some inconsistencies. It is a review article which, with more recent information, has become scientifically useful. However, I would have liked it to take a closer look at the influence on consumers and the public health strategies involved. However, it is in a position to be published. I congratulate the authors.

Author Response

Dear Editors,

We would like to thank you for the timely work carried out in the management of the submitted paper. Please find enclosed the revised version of the manuscript foods-2785866. The last comment by reviewer n.4 has been considered and the manuscript has been revised accordingly.

I hope you will find this review suitable for publication in the Foods Special Issue entitled: “Strategies to Improve the Functional Value of Meat and Meat Products”.

Sincerely yours,

Dr. Marco Iammarino on behalf of co-authors

Reviewer 4 comment:

The article has been restructured to give it more scientific consistency and to correct some inconsistencies. It is a review article which, with more recent information, has become scientifically useful. However, I would have liked it to take a closer look at the influence on consumers and the public health strategies involved. However, it is in a position to be published. I congratulate the authors.

Answer: Thanks for your comments. Following the last reviewer’s suggestion, a new comment has been added at lines 513-515.
